# How Football Players’ Age Affect Passing Patterns of Play According to Field Location

**DOI:** 10.3390/children10010157

**Published:** 2023-01-13

**Authors:** Nuno Coito, Hugo Folgado, Diogo Monteiro, Bruno Travassos

**Affiliations:** 1Department of Sport Sciences, Universidade da Beira Interior, 6201-001 Covilhã, Portugal; 2Escola Superior de Desporto Rio Maior (ESDRM-IPSantarém), 2040-413 Rio Maior, Portugal; 3Life Quality Research Centre (CIEQV), 2040-413 Rio Maior, Portugal; 4Departamento de Desporto e Saúde, Escola de Saúde e Desenvolvimento Humano, Universidade de Évora, 7000-727 Evora, Portugal; 5Comprehensive Health Research Centre (CHRC), 7002-554 Evora, Portugal; 6ESECS—Polytechnic of Leiria, 2411-901 Leiria, Portugal; 7Research Center in Sport Sciences, Health Sciences and Human Development (CIDESD), 5001-801 VilaReal, Portugal; 8Portuguese Football Federation, 1495-433 Cruz Quebrada-Dafundo, Portugal

**Keywords:** football, age, field zone, passing, receive

## Abstract

This study aimed to characterize the passing patterns that support collective tactical behaviour in football players of different ages (U15, U17, and U19) in different field zones. Two hundred and twenty-eight male players, divided into U15, U17, and U19, participated in the study. Cluster analysis was used to group the passes into three sizes (short, medium, and long). The chi-square test was used to analyse the effect of player age on game-passing patterns in each field zone. The results revealed that long and medium passes were used more in areas close to the goals and short passes in the middle area of the field, concerning all ages (*p* < 0.001). Furthermore, the analysis of the relative distance between the ball carrier and the receiver indicated that older players (U17 and U19) used more distant players to pass the ball in medium and long passes. These results can help coaches design small-sided games according to the players’ ages and adjust to the field’s space and the numerical relationship, thus creating a greater transfer from training to competition.

## 1. Introduction

Over the years, football has revealed a great evolution concerning its laws, tactical systems, styles of play used, and, particularly, training methods. One of the reasons for this development concerns the increased use of technologies that allow a new understanding of the game [1,2] and a better explanation of the performance factors (physical, technical, and tactical) that characterize teams and players [3,4]. Likewise, scientific research in football has improved the capacity to explain the game [5] and understand the impact of contextual variables (e.g., venue, match status, quality of opposition, and match period) in the style of play and the tactical behaviour of the teams [6,7]. Such knowledge allow coaches to identify the important aspects that support the design of training sessions and ensure a better transfer of practices to competition, i.e., improve the performance of their teams and players [8].

For example, the analysis of positional data, passing patterns of play, or network analysis allow some improvements in the tactical exploration of space–time relations in different game moments by different teams [9,10]. The tactical behaviours of players and teams seem to be identifiable and reveal a signature that is dependent on the relationship they establish between teammates, opponents, and the position of the ball in relation to the goal [11,12].

Moreover, previous studies have shown that, depending on the zone of the field where the ball is located, the players’ tactical behaviours vary, causing changes in the collective dynamics between teams [13,14]. Likewise, in the defensive sector, defenders tend to show low variability in displacements and in individual tactical actions (number and type of passes). On the other hand, in the attacking sector, attackers tend to reveal variability in space–time relations to create imbalances in the opponent [15,16]. Moreover, midfielders tend to reveal a greater number of actions in different game contexts due to occupying a more central position on the field, with higher variability in players’ relations [17] from which a large part of attacks are built. Thus, in analysis of players, the tactical action must be contextualized based on the variables that constrain their possibilities of action, such as the area of the field, the number of players closest to the ball [18,19], or the location of the teammate who can receive the ball [20].

The way players explore the game’s action possibilities, and how the ball moves around the pitch is, however, not only based on the context of play but also on the players’ action capabilities [21,22]. For example, older players with more technical–tactical skills tend to play more in width than in depth, enhancing a more elaborate game, while younger players with less skill tend to use a more direct game [23] but with less effectiveness for progression or to create shooting situations [24]. Thus, older and more skilled players generally show a greater capacity to adapt to the existing playing space, allowing the emergence of more functional collective behaviours [25]. On the other hand, at the level of the ball patterns of play, younger players tend to present a random exploration of possibilities for action, with a dependency on passes around specific players and less efficiency in the passes of older players [26]. That is, younger players tend to be less intentional in exploring the free spaces and, at the same time, present less motor efficiency to perform with precision in the available space and time. Therefore, players’ age, or even the players’ maturation stage, can influence technical actions and motor skills [27], with implications in tactical behaviour and decision-making [28].

Summarizing, it may be considered that the patterns of passes allow the assessment of the offensive style of the teams as well as the teams’ qualitative level to adapt to the variation of the existing playing space. Thus, there is a need to compare different age groups to understand the variation of tactical behaviours through the passing pattern [29]. Therefore, this study aimed to characterize the passing patterns that support the collective tactical behaviour of football players of different ages (U15, U17, and U19) by identifying their length, field zone, and relative distance between the passer and the receiver. Variations in the passing patterns of play depending on the field zones where the ball was were expected. Moreover, identifiably different lengths of passing and the relative distance of the players that receive the ball, according to variations in players’ age, were expected.

## 2. Materials and Methods

### 2.1. Participants

A total of two hundred and twenty-eight male players who competed in the Portuguese national first division for each age group was represented by under 15 (*n* = 76 players, age 14.4 ± 0.4 years, height 1.61 ± 0.07 weight 52.2 ± 9.0, biological maturation 84% middle and 16% early), under 17 (*n* = 76 players, age 15.6 ± 0.5 years, height 1.74 ± 0.05, weight 63.1 ± 7.5, biological maturation 94% early and 6% middle), and under 19 (*n* = 76 players, age 17.7 ± 0.5 years, height 1.78 ± 0.09, weight 75.3 ± 9.3, biological maturation 100% early).Three different clubs participated in each age group.

### 2.2. Procedure

A total of nine games were played using the official rules (three by age).Moreover, biological maturation was calculated for boys using the maturity offset formula (MO) = −7.999994 + (0.0036124 × (age × height)); R^2^ = 0.896; SEE = 0.542 [30].

The coaches defined the team composition in each simulated game to ensure balanced and competitive matches. Positional data of all the players were collected using individual global positioning system [GPS] units at 10 Hz (S5, Catapult Innovations, Melbourne, Australia). Goalkeepers were not included in the study. GPSs were turned on 15 min before each game. The games were recorded using a camera (Panasonic HC-V160).

Notational analysis was performed by recording the following player actions: passing the ball, receiving the ball, shooting, recovering the ball, and fouls. For this analysis, the software LongoMatch (version 1.3.7., Barcelona, Spain) was used, considering the time of each action for synchronizing the ball events with the GPS positional data. In addition, a visual representation of each simulated match was processed, presenting the ball position and displacement. Finally, this representation was used for possible notational error correction by comparing it with the original video [31].

The field was identified into different six zones according to previous studies [13] (Figure 1). The position of the ball passer and ball receiver was classified according to those different zones of the field for the analysis of the passes pattern. Zone 1 (Z1) is located near the team’s own goal and zone 6 (Z6) is located near to the opponent’s goal.

Attacking field players’ distance to the ball carrier were also calculated, as well as their relative distance, ranked for every moment of the match, from the closest player (P1) to the furthest player (P9) [32].

### 2.3. Data Analysis

Through a cluster analysis, the passes (total = 4537; U15 = 1184; U17 = 1893; U19 = 1460) were grouped into long, medium, and short using the k-means method, computed per age group. Calculations were made to obtain the averages of the long passes (31.67 m ± 6.72), the medium passes (16.69 m ± 2.93), and the short passes (8.29 m ± 5.62). In order to compare the pass distribution, considering the pitch zones, the pass range, and the ages, the simple chi-square test was used (contingency table) with a defined significance level of *p* < 0.05. The adjusted residues were computed for pairwise comparison to reveal the differences within each group [33]. The pairwise *p*-value was calculated using the Bonferroni correction method. For statistical analysis, the jamovi project software was used (version 1.6).

## 3. Results

Table 1 reveals the mean value of long, medium, and short passes (mean ± standard deviation) in relation to players’ age and considering the six zones of the field. In general, in all the players’ age groups, the short passes turned out to be the most used (U15—44.7%; U17—49%; U19—45.3%), followed by medium (U15—39.3%; U17—37.9%; U19—40%) and, in a much lower quantity, long passes (U15—16%; U17—13%; U19—14.7%) (Table 1).Pairwise comparison revealed that U15 used more long passes than U17 and U19.

However, statistically significant differences were noticeable (χ² = 113.0, *p* > 0.001) in the occurrence of different types of passes according to the field zones. The significant variations were recorded in the long passes in zone 1 (*p* < 0.001), zone 3 (*p* = 0.000), and zone 6 (*p* = 0.000) and in the short passes in zone 1 (*p* = 0.000), zone 4 (*p* = 0.001), and zone 6 (*p* = 0.000).There were no significant differences in the medium passes. In the zones nearest to the goal (zones 1 and 6), medium passes were mostly used, while short passes were predominantly used in the medium zone of the pitch (zones 2 to 5). In the end, short passes occurred less in zone 1 and long passes in the other zones (Figure 2). Overall, the teams revealed a relatively higher frequency of medium and long passes than short passes closer to the goal.

Considering the variation in the type of passes according to the pitch zones by age group, the results revealed significant differences for all the age groups analysed (U15, χ² = 41.6, *p* < 0.001; U17, χ² = 39.9, *p* < 0.001; U19, χ² = 53.5, *p* < 0.001) and the types passes analysed (long, χ² = 18.7, *p* < 0.044; short, χ² = 20.1, *p* < 0.029), except for medium passes (χ² = 15.0, *p* = 0.132).

Moreover, in the distribution of passes by each zone, the results revealed significant differences regarding long and short passes in zone 6 (*p* = 0.000), with a value above that expected in the U15.

Significant differences appeared in long and short passes in zone 1 and 3 (*p* = 0.000), with a value above that expected for long and below for short passes in the U17. Significant differences were reported in long and short passes in zone 1 (*p* = 0.000), with values above the expected for long and below for short passes in the U19. In general, long passes exceeded expectations in the areas close to the goals and were below expectations in the middle areas of the field. In opposition, short passes were below those expected in the areas close to the goals and above the expected in the middle areas of the field.

Moreover, we observed significant differences in the players who received the ball considering age in long passes (χ² = 32.2, *p* = 0.010) and medium passes (χ² = 31.8, *p* = 0.011).

Concerning short passes, in all ages, the player who received the most passes was P1. In this type of pass, there were only differences in the distribution for the fifth player, with the U17 considering this player at this distance (Figure 3). Regarding medium passes, the player who received more passes in the U15 was P2 (*p* > 0.05); in the U17 (*p* = 0.000) and U19 (*p* = 0.000), it was P3. As for long passes, the player who received more passes in the U15 was P5 (*p* = 0.000); in the U17 (*p* = 0.000) and U19 (*p* = 0.000), the player who received more passes was P6. The results show a trend for the younger players to use relatively closer players for long and medium passes compared to older players.

## 4. Discussion

This study aimed to characterize the passing patterns that support the collective tactical behaviour of football players of different ages (U15, U17, and U19). According to our expectations, some variations in the passing patterns of play were observed according to the field zones of the ball where the pass was performed. For example, in all ages, the most used pass in the areas close to the goals was the medium pass, and in the medium zone, it was the short pass.Moreover, according to our expectations, variations in age category promoted variations in the type of most frequent passes and in the player that receives the ball.

### 4.1. Effect of Field Zones

In the analysis of the effect of the variation of the zones in which the ball was located, we verified that, in zone 1, medium passes were the most used. In line with previous studies, in zone 1, teams in possession of the ball tended to have numerical superiority, with great values of individual areas per player, regardless of age [17,18]. Therefore, in this play area, near the team’s own goal zone, the players needed to use medium and long passes that allowed the creation of space to start the offensive game safely and, consequently, to advance on the field and pass the opponent’s first defensive line.

In the middle area of the field (Z2 to Z5), there was an increase in short passes and a decrease in long passes. In these areas, the short pass was the most used, suggesting that teams, regardless of age, sought to establish functional relationships between the closest players to create a numerical advantage in the group relationships between the players closest to the ball [34] and, in this way, attempt to overcome direct opponents and create defensive imbalances for the opponent [9].

While zone 2 is characterized by the space where the organized attack begins, normally with little pressure on the ball and maintaining the numerical superiority of the attack, in zones 3 and 4, the individual playing area per player is smaller compared to the zones close to the goals [13,14], promoting more ball possession for the exploration and creation of progression possibilities [18] and more possibilities for shorter passes. As verified in previous studies, zones 2 to 5 correspond to the areas of the field with high values of pass variability in view of the space constraint to play and the plasticity in the offensive tactical behaviour necessary to promote progression in the game area [35].

In zone 6, with the decrease in distance to the opponent’s goal, the most used type of pass was the medium pass due to the change concerning the objectives of ball circulation [creating imbalances to finish] and the emergence of possibilities to shoot on goal [11]. In this area, there tends to be a lower number of passes compared to the middle area of the field due to the proximity of the opposing goal that motivates the dribble and the shot. In this area, the team in possession of the ball tends to be outnumbered, and the short distance to the defending goal enhances the emergence of individual actions on the ball at greater risk [36]. The goal’s location proved to be an informational invariant that conditioned the players’ interpersonal relationships and individual and collective behaviours [11,12]. The field areas motivate the players’ need to explore different possibilities of action, so as to discover the best solution according to the game requirements [23].

### 4.2. Effect of Players Age on Field Zone Dynamics

The effect of age on the different types of passes showed significant differences in long and short passes. The results revealed that younger players (U15) tend to use long (Z6) and medium (Z5 and Z6) passes more than older players (U17 and U19). This evidence is in line with previous studies, which revealed that younger players tend to adopt behaviours that allow them to approach the opponent’s goal faster compared to older players [37].

On the contrary, U19 and U17 tended to use more long, medium, and short passes in zone 4 and zone 5 compared to U15. Previous studies suggest shorter distances between strikers and defenders in the middle zone of the field [38]. This spatial–temporal decrease potentiates variability in the possibilities for action depending on the variability of the spatial occupation of teammates and opponents [39], leading to the emergence of greater variability in the types of passes in order to obtain more functional actions adjusted to the conditions encountered [40].

The data suggest that older players showed greater variability in passes and showed greater adaptability in smaller playing areas [37]. In line with other studies, older players revealed a greater adaptability to the effects of manipulating the playing areas compared to younger players [21,41]. Thus, the greater game experience and consequent ability to functionally explore possibilities for action [42] of older players makes them more efficient in passing and gives them a better occupation of field spaces to receive the ball [39]. Our data also suggest that the players who most often received the ball were the closest to the carrier in the U15, regardless of the pass type. On the other hand, older players passed the ball more often to teammates, with more distant relative positions according to the pass length. For example, in the long pass, the player who received the ball most was P6 in U17 and U19. In the medium pass, it was P3 in the U19, and, with similar values, it was P2 and P3 in the U17, while, in short passes, the players closest to the ball (P1 and P2) revealed a greater tendency to receive the ball, regardless of age. Older players’ ability to pass to more distant players may be due to their apparent advantages in body mass [44] and level of maturity compared [43,44] to younger players. The data suggest that the U15 made more long passes to the same relative position of the receiver compared to the U17 and U19, which is in line with another study that showed higher values of individual areas per player in the U15 level [37].

Moreover, older players with more playing experience are more flexible to the game dynamics [25,40], while younger players tend to play with more rigid behaviours [26]. However, greater knowledge of the game allows players to gather more and better information from the environment, perceiving more possibilities for action [44]. The study suggests that the sizes of passes performed in the soccer game were influenced according to the field area and player’s age. In practical terms, these results reinforce the fact that the creation of exercises should consider the variation of the field location and the use of different numerical relationships in order to achieve a greater adjustment to the game conditions and ensure greater transfer between training and competition [16].

Due to the obtained results, and in order to better prepare the players to explore the best passing actions in zone 1, we suggest that the design of the should include the numerical superiority of attacking team [17], with more than four players in the defence, in order to clearly define the first defensive line and space to explore in-between lines [45]. The space used should be manipulated to allow players freedom of action in carrying medium passes, but also long and short ones according to the spatial–temporal relations with opponents. The manipulation of spaces should be considered according to the age of the players, with wider spaces for U15 than for U19 [37]. We also suggest that coaches could introduce two goals that the defensive team can use to score when recovering the ball to increase the feeling of danger in the case of losing ball possession [46].

Regarding zones 2 to 5, we suggest that the design of the SSCGs should include a numerical equality of teams from 5 × 5 to 8 × 8 [17,47], and promote variability in the exploration of short to medium passes both laterally and longitudinally. For that, the manipulation of spaces should be considered according to the age of the players and number of players involved, with wider spaces for U15 and U17 than for U19 [37]. The use of larger spaces and fewer players, particularly to youth players, tends to increase the number of individual actions in the exploration of possibilities for play due to the low restrictions in the spatial–temporal conditions [48]. We also suggest that coaches could manipulate the field such as the orientation (regular, square, triangle) [49], or even restrict the spaces that players could use in the field [29], to promote an adaptive capacity to explore the environment and explore the possibilities for short and long passes according to the conditions of the environment. Moreover, by considering the restriction of space and time to play and the large number of players involved in the zones 2 to 5, we suggest the manipulation of the number of touches as a constraint to improve players’ capacity to adjust the body position according to the game flow, constantly pick up the information from the environment before to have the ball possession [50], and promote the passing skill [51].

In zone 6, we suggest that the design of the SSCGs should include numerical relations up to 10 players (5 × 5) with a playing area that will enable possibilities of action in the variety of medium and long passes. In this area, we suggest that the design of the SSCGs should include numerical equality to promote the variability of short and medium passes and numerical superiority in the attacking team of two or more players in relation to the opponent to allow more shooting opportunities [32]. Moreover, the zone of the field and the manipulation of spaces should be considered according to the age of the players and number of players involved, with wider spaces for U15 than U17 and U19 [37]. We also suggest the use of several goals [three to six] to promote greater distances between the players of one’s own team and those of the opponent team in order to increase the variability of the individual and collective behaviours of the teams [46].

## 5. Conclusions

This study suggests, in line with the data from the game, the need to expose players to environmental conditions during training which are appropriate for the type of passes intended for the game. The ratio number of players and space motivates different ways of exploring the game depending on the players’ own abilities [52]. For this, in the design of small games, the coach must consider the players’ age to adjust accordingly to the game area and its proximity to the goal and the ratio number of players/game space.

Besides the study’s contribution for practical purposes, some limitations should be acknowledged. Firstly, the games were not official. Thus, it reduces the competitive levels of an official game, despite ensuring a controlled environment. In addition, the teams played according to their own game model without any control of playing formations used over the matches (e.g., Gk + 4 + 4 + 2 or Gk + 4 + 3 + 3).

Future studies may consider the directionality of passes to help understand the playing style of teams and relate it to the types of passes in the different offensive phases of the game. The distance between strikers and defenders should also be considered in future studies to understand the type of pass with the proximity of the defender in relation to the ball receiver. Moreover, further research should consider not only the age/category of players but also their maturation stage and the relationship with the type of passes performed. The maturation stage should be considered in all studies with youth players due to its implications in their characterization of capabilities for action.

## Figures and Tables

**Table 1 children-10-00157-t001:** Descriptive analysis of passes length by age.

Age	Zone	Passes Length					
		Short		Medium		Long	
		n (%)	m ± sd	n (%)	m ± sd	n (%)	m ± sd
U15	1	2.8	8.39 ± 2.63	3.9	18.49 ± 2.53	2.4	29.69 ± 5.00
	2	6.8	8.23 ± 2.54	6.1	16.93 ± 3.06	2.2	29.75 ± 8.44
	3	10.9	8.54 ± 2.44	9.8	16.37 ± 2.94	3.2	33.99 ± 8.67
	4	13.4	8.26 ± 2.61	8.3	16.55 ± 3.01	3.9	30.40 ± 6.76
	5	7.9	8.23 ± 2.55	6.7	16.68 ± 2.98	1.6	32.35 ± 10.14
	6	3.0	8.74 ± 3.00	4.6	16.96 ± 3.11	2.8	32.23 ± 7.62
	Total	44.7	8.36 ± 2.57	39.3	16.82 ± 3.00	16.0	31.56 ± 7.91
U17	1	2.4	8.14 ± 2.73	2.9	16.50 ± 2.85	2.1	39.12 ± 20.35
	2	8.6	8.19 ± 2.68	6.9	16.71 ± 2.80	2.0	30.32 ± 9.46
	3	12.6	7.96 ± 2.66	9.1	16.48 ± 2.90	2.2	32.40 ± 7.96
	4	11.8	8.31 ± 2.50	7.8	16.47 ± 2.86	2.9	31.21 ± 7.26
	5	9.2	8.19 ± 2.62	6.8	16.19 ± 2.88	2.2	29.86 ± 5.81
	6	4.5	8.12 ± 2.94	4.4	16.79 ± 3.13	1.5	38.85 ± 21.25
	Total	49.0	8.15 ± 2.65	37.9	16.50 ± 2.89	13.0	33.04 ± 12.92
U19	1	1.6	8.37 ± 2.75	3.3	17.32 ± 2.48	2.2	31.58 ± 7.69
	2	8.3	8.36 ± 2.66	7.3	16.80 ± 2.82	2.3	29.39 ± 6.78
	3	11.6	8.30 ± 2.56	9.4	16.49 ± 2.78	2.6	30.80 ± 5.76
	4	13.7	8.24 ± 2.60	10.6	16.38 ± 2.87	3.9	29.99 ± 6.31
	5	7.6	8.40 ± 2.68	6.0	16.75 ± 3.19	2.0	28.90 ± 5.37
	6	2.5	9.18 ± 2.33	3.4	17.53 ± 2.95	1.7	32.60 ± 7.47
	Total	45.3	8.37 ± 2.60	40.0	16.76 ± 2.87	14.7	30.42 ± 6.57

n: relative frequency; m ± sd: Mean ± standard deviation.

**Figure 1 children-10-00157-f001:**
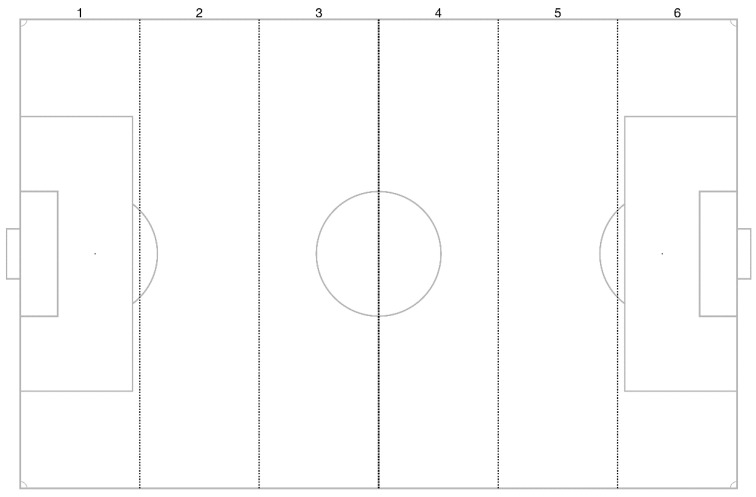
The field was divided into different six zones.

**Figure 2 children-10-00157-f002:**
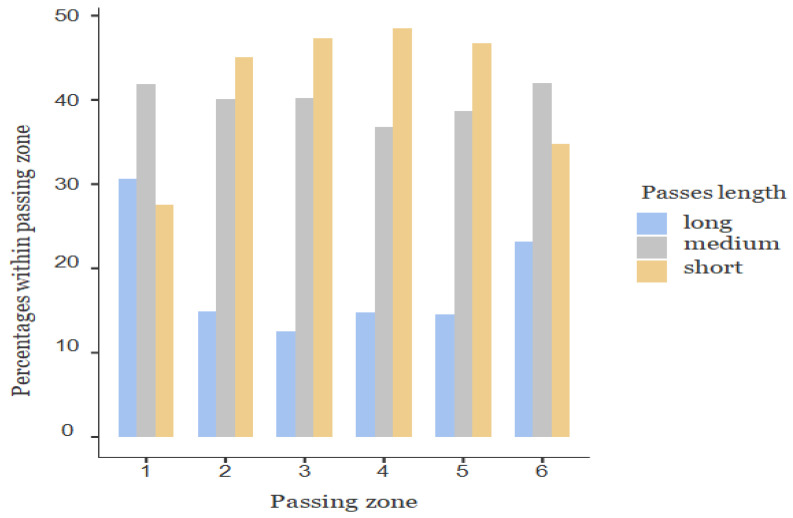
Distribution of the total number of passes by field zone.

**Figure 3 children-10-00157-f003:**
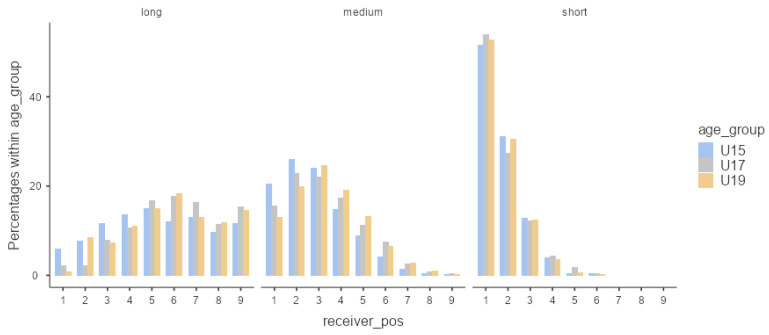
Passing distribution relative frequency of completed passes considering the relative distance between the ball receiver and the ball carrier, per age group and pass length.

## Data Availability

Data regarding this submission can be provided upon reasonable request to the primary author.

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
