# Peer review of "How Football Players’ Age Affect Passing Patterns of Play According to Field Location"

_children, 2023, doi:10.3390/children10010157_

Round 1
Reviewer 1 Report
The manuscript is interesting, and the goal is extremely important for soccer practice. However, the authors did not consider the effect of biological maturation! Maturation is associated with lower limb power in soccer players and influences the increase of anaerobic performance which can interfere with the passing distances (short, medium and long) and can interfere with the motor behavior pattern of soccer players. The authors report stature and chronological age data, through them it is possible to calculate the stages of biological maturation by the method proposed by Moore et al, (2015), I suggest that the authors calculate the maturation stages, indicate how many athletes of each maturation stage exist per group (categorical variable: late, middle and early) and check if maturation (scalar variable values <-1 to >1) is correlated with pass distances in each category, then check by X2 considering biological maturation stage instead of chronological age. Adding these analyses will answer whether this movement pattern stems from age category or biological maturation stage. By this adjustment, the authors will have to add in the introduction information about the influence of biological maturation on soccer players' performance. Please take a look at the following study:
1) Moore, S. A., McKay, H. A., Macdonald, H., Nettlefold, L., Baxter-Jones, A. D., Cameron, N., & Brasher, P. M. (2015). Enhancing a somatic maturity prediction model. Medicine & Science in Sports & Exercise, 47(8), 1755-1764. DOI: 10.1249/MSS.0000000000000588.
2) Abbas Asadi, Rodrigo Ramirez-Campillo, Hamid Arazi & Eduardo Sáez de Villarreal (2018): The effects of maturation on jumping ability and sprint adaptations to plyometric training in youth soccer players, Journal of Sports Sciences, DOI: 10.1080/02640414.2018.1459151
3) Kaczor, J. J., Ziolkowski, W., Popinigis, J., & Tarnopolsky, M. A., (2005). Anaerobic and aerobic enzyme activities in human skeletal muscle from children and adults. Pediatric Research. 57(3), 331-335. doi: 10.1203/01.PDR.0000150799.77094.DE
Author Response
R: Thank you for the suggestion! It is really an approach that should be followed in further research. We have calculated and reported in the manuscript the maturation stages of the players of each age. The U15 were 84% middle and 16% early stage, the U17 were 94% early and 6% middle and the U19 were 100% early. It means that there was a predominance in U15 to middle stage and in U17 and U19 to early. This results fit very well with the differences in the passing distance as suggested. We have added some information to stress this relationship.
Unfortunately, we don’t have possibility to calculate the correlation between the stage of maturation of each player and the distance of passes. Our data collection was only registered based on the distance of passes per age category. We didn’t register the player that made the pass.
However, agreeing with the suggestion made by the reviewer, we have stressed at the end of the manuscript that future research should consider the maturation stage instead of the only the age category.
Reviewer 2 Report
Dear Authors,
the paper sound interesting but have some structural errors, please check carefully all paper. Further, I recommend that you use a native speaker to check English throughout the paper
ABSTRACT
1. Please specify the sex of subjects
INTRODUCTION
2. Please expand the concept in line 34 -37
3. Please explain better the concept in line 62 – 70, specifying the reasons
4. Please rephrase the aims of the study, so they are not very clear
METHODS
5. Line 85. Please specify the level
6. Line 91. Double point: Began..
7. Line 95-96. Which is the methods for dividing the teams? What’s mean homogeneous? Please explain
8. Line 108-115. Different line spacing, please correct
9. Line 117. Please correct the reference
10. Line 122. Double space, please correct “furthest player”
11. Please rewrite the data analysis section in more clear manner
RESULTS
12. TABLE 1. Please insert the abbreviation in the caption (M, SD, etc.). further, write all the letters of line one (m, sd, n) on the same plane
13. Please insert χ2 in all results
14. Please check carefully the results, randomized the p with two or three number after point
15. Please correct the p in line 152, replacing the comma with the dot
16. Insert significance in figure 2
CONCLUSION
17.
18. Please insert practical application and limit of the study
19. Check the author contribution
REFERENCE
20. Please check the reference, and correct carefully in accord with the guidelines of the journal:
11. Headrick J, Davids K, Renshaw I, Araújo D, Passos P, Fernandes O. Proximity-to-goal as a constraint on patterns of behaviour 364 in attacker–defender dyads in team games. Journal of Sports Sciences. 2012;30[3]:247-53.
Could be:
11. Headrick, J.; Davids, K.; Renshaw, I.; Araújo, D.; Passos, P.; Fernandes, O. Proximity-to-goal as a constraint on patterns of behaviour in attacker-defender dyads in team games. J Sports Sci. 2012, 30(3), 247-53.
Author Response
ABSTRACT
- Please specify the sex of subjects
R: Changed accordingly. Please see L22
INTRODUCTION
- Please expand the concept in line 34 -37
R: Changed accordingly. Please see L38-44
- Please explain better the concept in line 62 – 70, specifying the reasons
We have stressed the relationship between players capabilities and exploration of possibilities for action. Please see L74-78
- Please rephrase the aims of the study, so they are not very clear
R: Changed accordingly. Please see L83-90
METHODS
- Line 85. Please specify the level
R: Thank you for your comments.Please see L95
- Line 91. Double point: Began..
R: Thank you for your comments.Please see L102
- Line 95-96. Which is the methods for dividing the teams? What’s mean homogeneous? Please explain
R: Thank you for your comments.Please see L109-110
- Line 108-115. Different line spacing, please correct
R: R: Thank you for your comments.Please see L122-130
- Line 117. Please correct the reference
R: Thank you for your comments.Please see L132
- Line 122. Double space, please correct “furthest player”
R: Thank you for your comments.Please see L138
- Please rewrite the data analysis section in more clear manner
R: Thank you for your comments.Please see L144 and L150
RESULTS
- TABLE 1. Please insert the abbreviation in the caption (M, SD, etc.). further, write all the letters of line one (m, sd, n) on the same plane
R: Thank you for your comments. Changed accordingly with suggestions
- Please insert χ2 in all results
R: We added all the χ2 in the results section
- Please check carefully the results, randomized the p with two or three number after point
R: Thank you for your comments. Changed accordingly with suggestions p with three number after point
- Please correct the p in line 152, replacing the comma with the dot
R: Thank you for your comments.Please see L152
- Insert significance in figure 2
R: Given the nature of the statistical testing, and of the data, we opted to present the statistical significance within the text, for readability. For figure 2, the statistical significanceis presented in lines 189 to 192.
CONCLUSION
- Please insert practical application and limit of the study
R: We added the limitations of the study, the practical implications and future research
- Check the author contribution
R: Thank you for your comments.Please see L374-377
REFERENCE
- Please check the reference, and correct carefully in accord with the guidelines of the journal:
- Headrick J, Davids K, Renshaw I, Araújo D, Passos P, Fernandes O. Proximity-to-goal as a constraint on patterns of behaviour in attacker–defender dyads in team games. Journal of Sports Sciences. 2012;30[3]:247-53.
Could be:
- Headrick, J.; Davids, K.; Renshaw, I.; Araújo, D.; Passos, P.; Fernandes, O. Proximity-to-goal as a constraint on patterns of behaviour in attacker-defender dyads in team games. J Sports Sci.2012, 30(3), 247-53.
R: Thank you for your comments.Please see L410
Round 2
Reviewer 1 Report
After the adjustments made by the authors, I suggest the acceptance of the manuscript!